**Data Availability Statement:** All relevant data are within the paper and its Supporting information files.

**Funding:** The author(s) received no specific funding for this work.

# Stiffness reduction and collagenase resistance of aging lungs measured using scanning acoustic microscopy

**Katsutoshi Miura** [ORCID] *

Department of Regenerative and Infectious Pathology, Hamamatsu University School of Medicine, Hamamatsu, Japan

* kmiura.hama.med@gmail.com

## Abstract

Lung tissue stiffness is altered with aging. Quantitatively evaluating lung function is difficult using a light microscope (LM) alone. Scanning acoustic microscope (SAM) calculates the speed-of-sound (SOS) using sections to obtain histological images by plotting SOS values on the screen. As SOS is positively correlated with stiffness, SAM has a superior characteristic of simultaneously evaluating tissue stiffness and structure. SOS images of healthy bronchioles, arterioles, and alveoli were compared among young, middle-aged, and old lung sections. Formalin-fixed, paraffin-embedded (FFPE) sections consistently exhibited relatively higher SOS values than fresh-frozen sections, indicating that FFPE became stiffer but retained the relative stiffness reflecting fresh samples. All lung components exhibited gradually declining SOS values with aging and were associated with structural alterations such as loss of smooth muscles, collagen, and elastic fibers. Moreover, reaction to collagenase digestion resulted in decreased SOS values. SOS values of all components were significantly reduced in young and middle-aged groups, whereas no significant reduction was observed in the old group. Protease damage in the absence of regeneration or loss of elastic components was present in old lungs, which exbited dilated bronchioles and alveoli. Aging lungs gradually lose stiffness with decreasing structural components without exposure to specific insults such as inflammation.

## Introduction

Lung aging is associated with a progressive tissue breakdown and a consecutive decline of respiratory function with structural remodeling [1,2]. For example, enlarged air space decreased alveolar stiffness, and increased functional residual capacity (the air volume remaining at the end of expiration) were observed in old lungs. Moreover, the FEV1/FVC ratio (the proportion of an individual's vital capacity that can expire in the first second of forced expiration to the full, forced vital capacity) decreased [2]. Lungs are exposed to many kinds of damages, such as ozone, aerosols, infection, allergens, and pollutants, and required tissue repair. Ineffective repair can lead to fibrosis/scarring or remodeling (e.g., emphysema and chronic

**Competing interests:** The authors have declared that no competing interests exist.

obstructive pulmonary disease). Fibroblasts secrete collagen and extracellular matrix proteins contributing to the stiffness, and the breakdown of these proteins causes airspace enlargement [3]. In disease conditions or experimental models of injury, fibroblasts from older individuals or mice produce more collagen, fibronectin, and matrix metalloproteinases, contributing to disease severity [2]. Although various insults induce interstitial fibrosis to cause stiffness, the presence and extent of alterations in the normal lung in the absence of remarkable inflammation remain unclear. Moreover, whether the alterations in the number of collagen fibers persist permanently or gradually resolve remains unclear. Observing structural changes simultaneously with mechanical alterations should aid in addressing these enduring research questions.

The standard light microscope (LM) only displays histological structure and follows qualitative changes. Scanning acoustic microscopy (SAM) shows histological and mechanical properties of tissues using the same slide observed through the LM [4]. SAM calculates the speed-of-sound (SOS) for each section to plot values that make histological images. SOS values correspond to tissue stiffness as higher speed correlates with more stiffness [5]. Moreover, protease treatment breaks the structural tissue components to reduce stiffness [6–8]. The sensitivity of lung tissues to protease digestion, which invariably occurs due to environmental damages or inflammation, can be evaluated by reducing SOS values after digestion. In this study, formalin-fixed, paraffin-embedded (FFPE) sections for pathological diagnosis, including young, middle-aged, and elderly patients, compared SOS values and sensitivity to collagenase digestion. In the present study, we examined normal lung aging through SAM.

## Materials and methods

### Participants and ethics

The study protocol conformed to the Declaration of Helsinki ethical guidelines. The Clinical Research Review Board of Hamamatsu University School of Medicine approved this study (approval no. 19–180). The study used pathology samples collected during routine surgery, and verbal and written informed consents were obtained from each patient before sample collection. The name and other identifying information were not used in the study. All procedures were conducted according to the approved guidelines and regulations of the Clinical Research Review Board of Hamamatsu University School of Medicine.

### Sample preparation

All lung specimens were obtained from 12 patients, including 7 male and 5 female patients, aged between 17 and 87 years who underwent partial lung resection in the Hamamatsu University Hospital in Japan between January 1, 2016 and December 31, 2018. The study cohort was divided into young (age, 10s to 20s), middle-aged (40s), and old (80s) groups (Table 1). Pathological diagnoses included spontaneous pneumothorax or localized carcinoma, and none of the lungs included secondary lesions. Normal lung sections away from the primary lesion were used for the present study, with patient consent.

### Comparison among fresh-frozen, formalin-fixed, and FFPE sections

To compare the impact of different fixation and embedding methods on results, FFPE and fresh-frozen sections were prepared from the same specimen collected from the lung of a 75-year-old woman. For fresh-frozen sections, a piece of the lung specimen was embedded in Tissue-Plus Optimal Cutting Temperature Compound (Fisher HealthCare, Waltham, MA, US), frozen in n-hexane at −80˚C, and cut in 10-μm-thick sections using a cryotome cryostat

**Table 1. List of human lung tissues used in the present study***.

| Group | Age | Sex | Pathological diagnosis |
|---|---|---|---|
| **Young** | **17** | **M** | **Spontaneous pneumothorax** |
| | **17** | **M** | **Spontaneous pneumothorax** |
| | **18** | **M** | **Spontaneous pneumothorax** |
| | **23** | **M** | **Spontaneous pneumothorax** |
| **Middle-aged** | **43** | **F** | **Adenocarcinoma, localized** |
| | **43** | **F** | **Adenocarcinoma, localized** |
| | **44** | **M** | **Adenocarcinoma, localized** |
| | **44** | **F** | **Adenocarcinoma, localized** |
| **Old** | **82** | **M** | **Adenocarcinoma, localized** |
| | **83** | **F** | **Pleomorphic carcinoma, localized** |
| | **83** | **F** | **Adenocarcinoma, localized** |
| | **87** | **M** | **Adenocarcinoma, localized** |

*All patients underwent partial lung resection and did not have serious complications or secondary lesions.

(Leica Biosystems, Tokyo, Japan). The sections were mounted on MAS-coated glass slides (Matsunami Glass, Kishiwada, Osaka, Japan), dried, and stored at −80˚C until use. The section without fixation was reserved as an unfixed/frozen section and soaked in physiologic saline before observation. Another section was fixed in 10% buffered formalin for 24 h at room temperature and washed in distilled water for SAM observation. For FFPE sections, the lung specimen was fixed in 10% buffered formalin for 24 h at room temperature, embedded in paraffin, and cut in 10-μm-thick sections. The sections were dewaxed in xylene, soaked in gradually reducing ethanol concentrations, and washed in distilled water before observation. All FFPE sections for all clincal samples were prepared using the same method.

To evaluate changes occurring with aging, the FFPE sections of specimens collected from different age groups were compared. The portions containing bronchioles, which were accompanied by arterioles and alveoli, were scanned to obtain SOS images. Sections with focal defects or uneven surfaces were omitted from measurements.

## SAM observations

Lung specimens were evaluated using a SAM system (AMS-50AI; Honda Electronics, Toyohashi, Aichi, Japan) with a central frequency of 320 MHz and a lateral resolution of 3.8 μm [9], as previously reported [8]. The transducer was excited with a 2-ns electrical pulse to emit an acoustic pulse [10]. Samples were placed on the transducer, and distilled water was used as coupling fluid between the transducer and the specimen. The transducer was used for both signal transmission and reception. Waveforms reflected from the surface and bottom of the sample were compared to measure each point's SOS and thickness. The waveform from a glass surface served as the reference value of SOS at 1485 m/s only through water.

The specimens were examined using the previously reported method [8,9], without modifications. Briefly, the mechanical scanner was arranged so that the ultrasonic beam was over the specimen to provide the SOS value of each point. The distance between the transducer and the specimen was adjusted to correctly determine the pulse wave. The scan width and line were 2.4, 1.2, and 0.6 mm$^2$. The sampling points were 300 in one scanning line and width, and each square frame comprised 300 × 300 points. Data from four scans were averaged to determine the value of each scan point to decrease noise interference. Mean SOS value of each component was calculated from five separate sections. To determine the SOS values, the values of a

small spot on the region of interest in the screen were recorded. For the evaluation of bronchioles and arterioles, measrements were obtained from at least five points across the entire wall circumference.

The points of interest were randomly selected from cross points on the lattice screen [11]. The region in SOS images, typically with a size of 1.2 or 0.6 mm$^2$, was chosen from the corresponding LM images.

## LM observation

For comparison, the same or nearby SAM sections were stained with hematoxylin and eosin and Verhoeff's elastic and Masson's trichrome stains. Using elastic and Masson's trichrome stains, collagen and elastic fibers were observed in blue and black colors, respectively.

## Immunohistochemical analysis

Immunostaining was performed using a commercially available DAKO EnVision™ System, Peroxidase kit (Dako, Glostrup, Denmark). Primary antibodies used to detect smooth muscle actin (SMA), type 1 collagen, and type 3 collagen were anti-SMA (ab5694, Abcam, Tokyo, Japan; 1:400), anti-collagen I (ab88147, Abcam, 1:100), and anti-collagen III (ab7778, Abcam, 1:1000), respectively. Heat-mediated antigen retrieval (95˚C, 20 min) was performed using a buffer balanced to either pH 6.0 (anti-SMA, anti-collagen III) or pH 9.0 (anti-collagen I) before staining. Next, the sections were washed three times with phosphate-buffered saline (PBS) and incubated with the primary antibody for 30 min at room temperature. Next, the sections were rewashed three times with PBS and incubated with the labeled polymer for 30 min at room temperature. The sections were washed three times in tris-buffered saline (pH 7.4) and incubated with 3,3′-diaminobenzidine as substrate chromogen for 10 min to reveal brown-colored precipitate at the antigen site.

## Collagenase digestion

Lung specimens of the young, middle-aged, and old groups were compared. Dewaxed paraffin sections were soaked in distilled water and submerged into a PBS solution containing 0.5-mM calcium chloride (pH 7.4) and 250 units/mL type 3 collagenase (Worthington, Lakewood, NJ, USA) at 37˚C for 1.5 h or 3 h [6]. The collagenase used in the present study was chosen for its substrate specificity to collagens with lower proteolytic activity compared with other collagenases. Digested sections were washed with distilled water before the SAM observation. The same sections were measured at 1.5 h and 3 h after digestion.

## Statistical analyses

Mean SOS values were calculated from at least five areas per structure (arterioles, bronchioles, and alveoli). One-way analysis of variance (ANOVA) with the Tukey–Kramer post hoc test was conducted to examine fixation and embedding effects, stiffness alteration in each component with aging, and collagenase effects on each component by aging. Three structures (arterioles, bronchioles, and alveoli) and different age groups (young, middle-aged, and old) were compared. Multiple comparisons with the Tukey–Kramer test were used to evaluate whether changes within each independent variable were significant. The manufacturer software for SAM (LabView 2012, National Instruments, Austin, TX, USA) and a commercial statistics software (BellCurve for Excel; Social Survey Research Information, Tokyo, Japan) were used to calculate areas-of-interest values, prepare dot blot graphs, and analyze ANOVA and Tukey–Kramer post hoc test results.

Before performing the Tukey–Kramer test, all datasets were examined for normal distribution and homogeneity of variance using Bartlett or Levine test. If the variances were equal, an ANOVA table was prepared to investigate the presence of different mean values among groups. In the presence of different means, the Tukey–Kramer test was used to determine groups with significantly different mean values. In the case of non-equal variances, a generalization of 2-sample Welch or Brown-Forsythe test was used. A $p$ value of $<0.05$ was considered to indicate statistically significant for all analyses.

## Results

### SOS variation among different fixation methods

Fig 1 shows the SOS images of unfixed/frozen, 24-h formalin-fixed, and FFPE sections for comparison. The unfixed/frozen and formalin-fixed sections represented the same area, whereas the FFPE section was prepared from another area. The color alteration in the sections prepared using three different methods showed that all lung components exhibited gradually increasing SOS values, lowest observed in fresh-frozen sections, followed by formalin-fixed and FFPE sections. With all fixation methods, the arterioles including smooth muscle exhibited the highest SOS values, the bronchioles including collagen fibers exhibited intermediate SOS values, and the alveoli displayed the lowest SOS values.

The mean SOS values (± standard deviation) were significantly different among the bronchioles (Fig 2a), arterioles (Fig 2b), and alveoli (Fig 2c) using one-way ANOVA with the

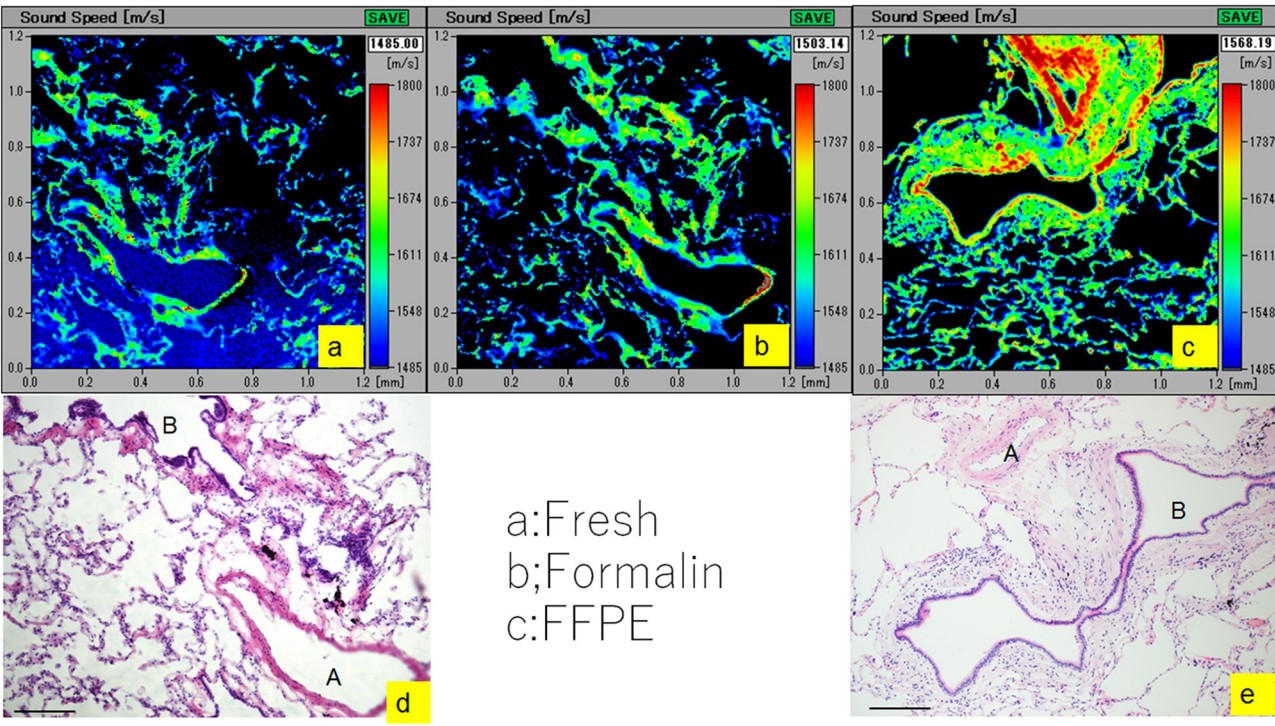

**Fig 1. Differences in SOS valuesamong fresh-frozen, formalin-fixed, and FFPE sections.** Comparison of SOS images of unfixed, fresh-frozen (**a**), formalin-fixed (**b**), and FFPE (**c**) sections. The corresponding light microscopic images of sections stained with hematoxylin and eosin (**d, e**). All images include bronchioles (B), arterioles (A), and alveoli. SOS values are represented by color indicators present on the right side of each image. The fresh-frozen section exhibits the lowest SOS values in bronchioles and arterioles, the formalin-fixed section displays intermediate SOS values, and the FFPE section exhibits the highest values. In alveoli, SOS values gradually increase from the fresh-frozen section to the formalin-fixed and FFPE sections. The blurred outline in the bronchiolar and alveolar walls has become clearer after fixation. Scale bar = 200 μm.

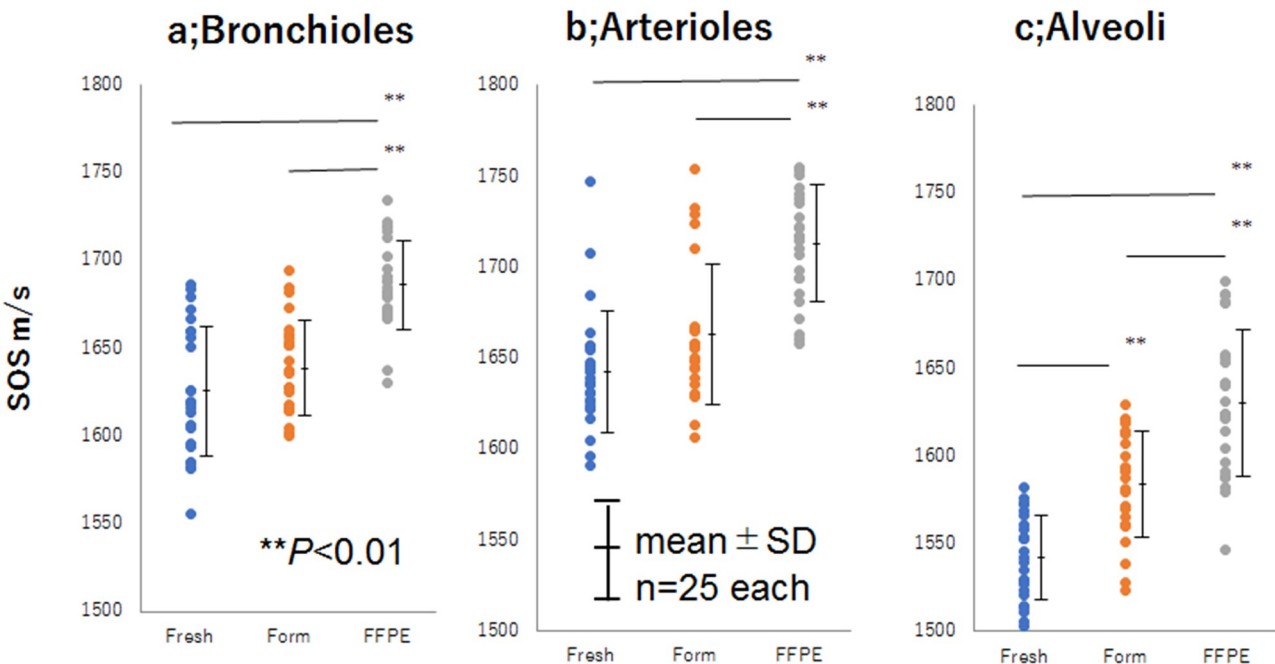

**Fig 2. Dot plots of SOS values among different fixation methods. (a)** Bronchioles, **(b)** arterioles, and **(c)** alveoli. Mean SOS values (± standard deviation) of bronchioles, arterioles, and alveoli were plotted to compare fixation and embedding effects. The FFPE sections exhibit significantly higher SOS values than fresh and formalin-fixed sections ($p < 0.001$).

Tukey–Kramer test (S1 Table, Fig 2). In all bronchioles, arterioles, and alveoli, statistically significant differences were observed between the fresh/unfixed and FFPE sections ($p < 0.001$) and between the formalin-fixed and FFPE sections ($p < 0.001$). Even in any FFPE section, the arterioles, bronchioles, and alveoli kept the highest, middle, lowest values in this order, respectively. These findings indicated that the FFPE tissue became stiffer but retained relative stiffness reflecting fresh samples.

## SOS values of the bronchioles, arterioles, and alveoli in different age groups

Fig 3 shows SOS images of the lung and the corresponding LM images in different age groups. SOS values of the bronchioles, arterioles, and alveoli were high in the younger group, which became lower with increasing age of the lungs. The young bronchiolar walls consisted of thick, blue-colored collagens in the LM image. Simultaneously, those of the old group composed a thin collagen layer with dilatation.

The arterioles were composed of thick, smooth muscles in the young lungs, whereas the middle-aged and old lungs showed arteriolosclerosis with thinner smooth muscles (Fig 4). The alveoli gradually increased in size with aging, and the walls varied in thickness with some splits found in an old lung. The breaks were found as artifacts made during the process because the old lung was mechanically less resistant.

One-way ANOVA showed a statistically significant difference in SOS values according to age groups (S2 Table). The post hoc Tukey–Kramer test showed that the SOS values of all three components (bronchioles, arterioles, and alveoli) of the young group were significantly higher than those of the middle-aged and old groups ($p < 0.001$ for both) (Fig 5). In all age groups, the SOS values were highest in arterioles, intermediate in bronchioles, and lowest in alveoli ($p < 0.01$).

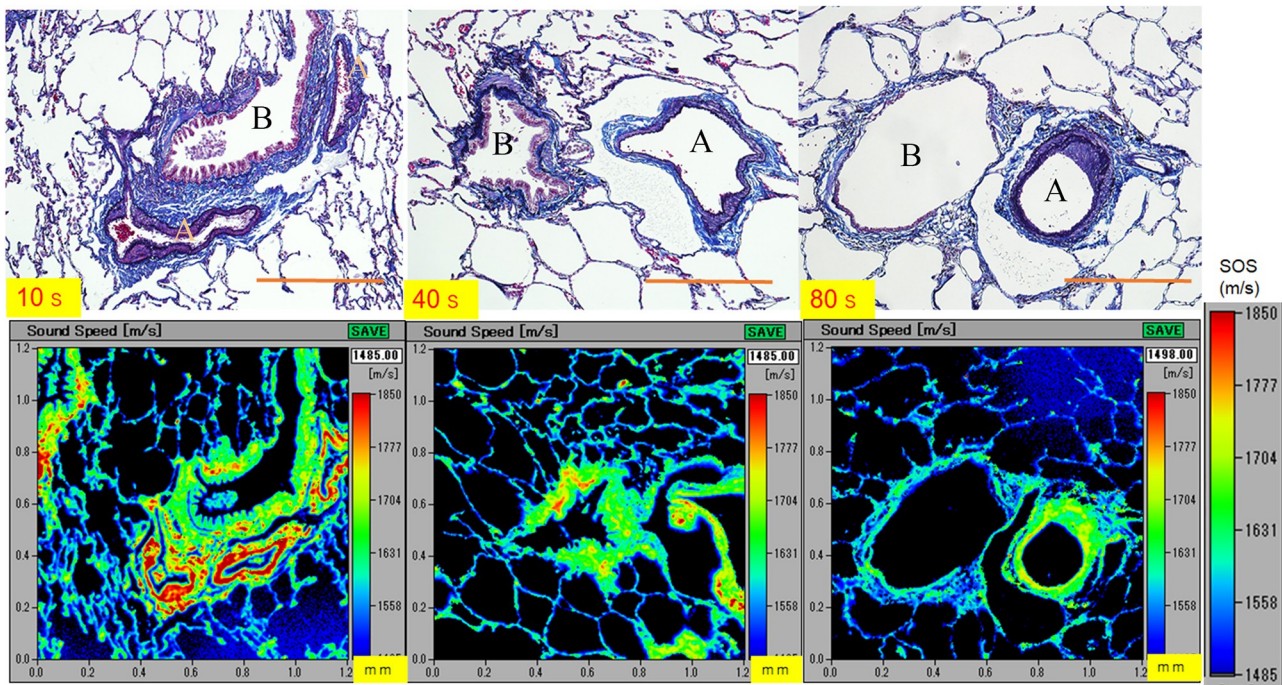

**Fig 3. SOS images of the young, middle-aged, and old lungs.** SOS values of the bronchioles, arterioles, and alveoli are high in the young group, which gradually become lower with the increasing age of the lungs, except for the intimal atheromatous region forming a ring shape in the arterioles of the old group, which exhibits high SOS values indicated as yellow. The corresponding LM images of sections stained with Verhoeff's elastic and Masson's trichrome. A, arteriole; B, bronchiole. Scale bar = 500 μm.

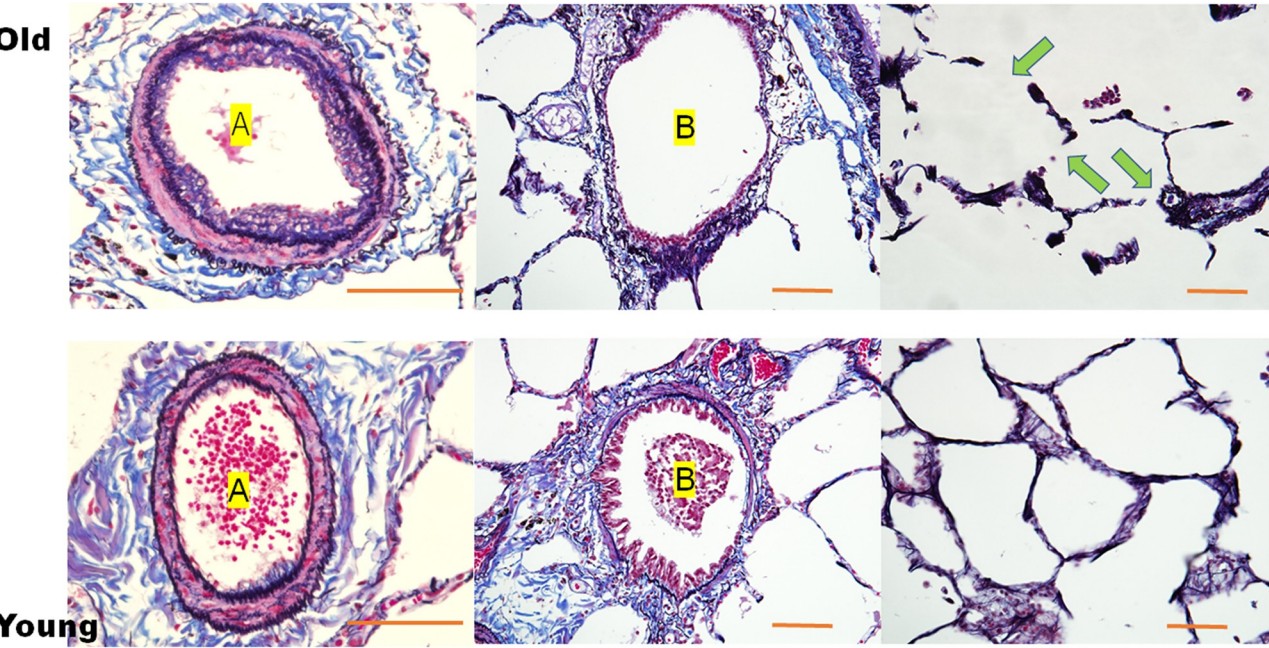

**Fig 4. LM images of sections stained with Verhoeff's elastic and Masson's trichrome stain.** In the old group (upper row), lungs are composed of arterioles (A) with a thin muscle layer and thick atheromatous intima, bronchioles with weak and poor collagen surroundings and sparse epithelial cells, and large alveoli composed of walls of varied thickness or disconnection (arrows). These splits are artifacts introduced during processing because of the lower mechanical resistance of old lung tissue. Conversely, young lungs (lower row) are composed of arterioles (A) with thick, smooth muscle; bronchioles (B) with concrete collagen surroundings and dense epithelial cells; and continuous alveolar walls with a regular size. Scale bar = 100 μm.

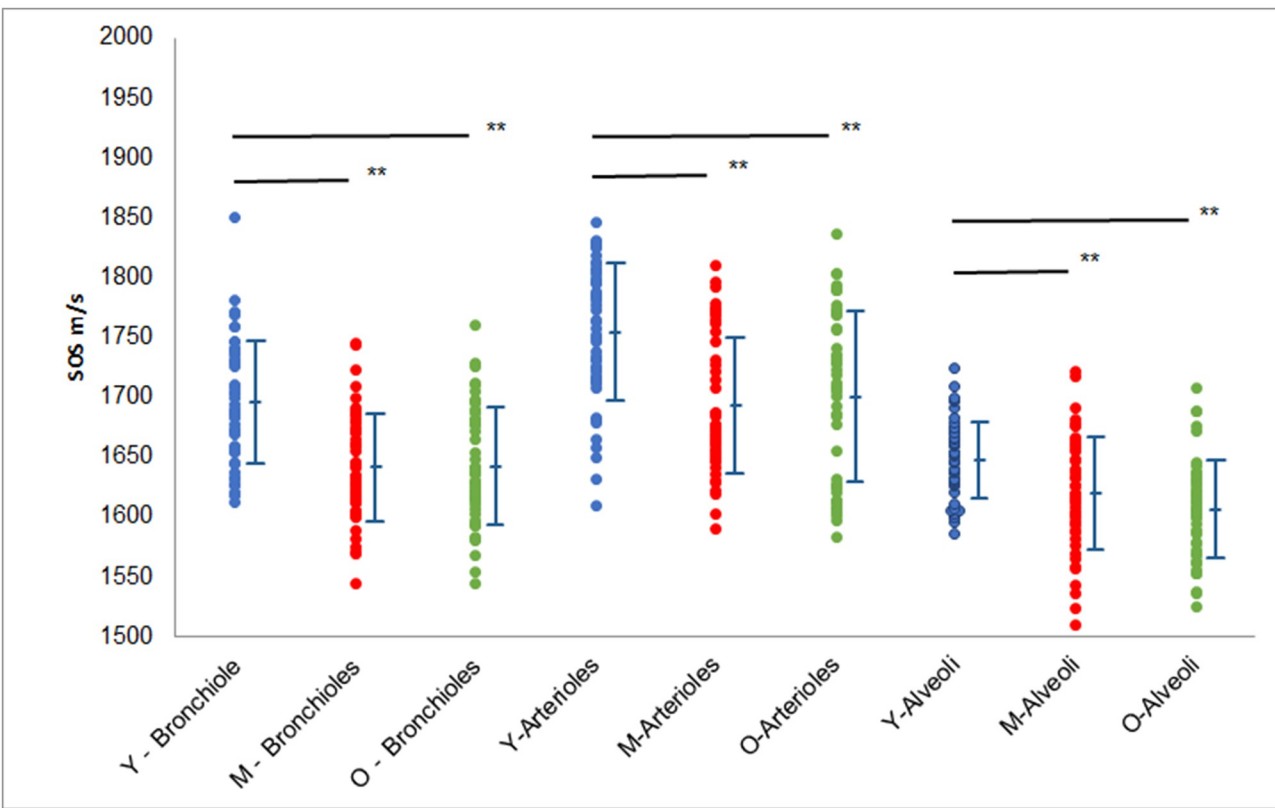

**Fig 5. Dot blots of SOS values in the bronchioles, arterioles, and alveoli among different age groups.** Mean SOS values (± standard deviation) of all three components are significantly highest in the young group compared with the middle-aged and old groups ($p < 0.001$). The mean SOS values of all three components are higher in the middle-aged group than in the old group, albeit without a statistical significant difference. In all age groups, the SOS values are highest in arterioles, intermediate in bronchioles, and lowest in alveoli. Y, young; M, middle-aged; O, old.

## Structure difference between young and old lungs according to immunostaining

Fig 6 shows immunostaining images of old and young lungs. Type 1 and type 3 collagen fibers surrounding the arterioles and bronchioles are thick in young lungs (Fig 6a and 6b). Conversely, an old lung exhibited thin faint collagen fibers around the arterioles and bronchioles (Fig 6d and 6e). The arteriole of an old lung consisted of double layers of collagen fibers. Some alveoli were composed of type 3 collagen extending from the bronchioles. Young lung had thick, smooth muscles comprising the arteriolar and bronchiolar walls (Fig 6c), whereas an old lung mostly lost smooth muscles (Fig 6f).

## SOS alteration after collagenase digestion

Young lungs (Fig 7) exhibited gradually decreased SOS values at the arteriolar and bronchiolar walls and alveoli, whereas old lungs exhibited nearly preserved SOS values after digestion (Fig 8). Blood contents within the arterioles, which served as an internal control for digestion, exhibited gradually declining SOS values after digestion.

In the time course of SOS values after collagenase digestion, one-way ANOVA for collagenase effects on changes in SOS values after digestion showed that all components, including arterioles, bronchioles, and alveoli, of young and middle-aged groups exhibited significant

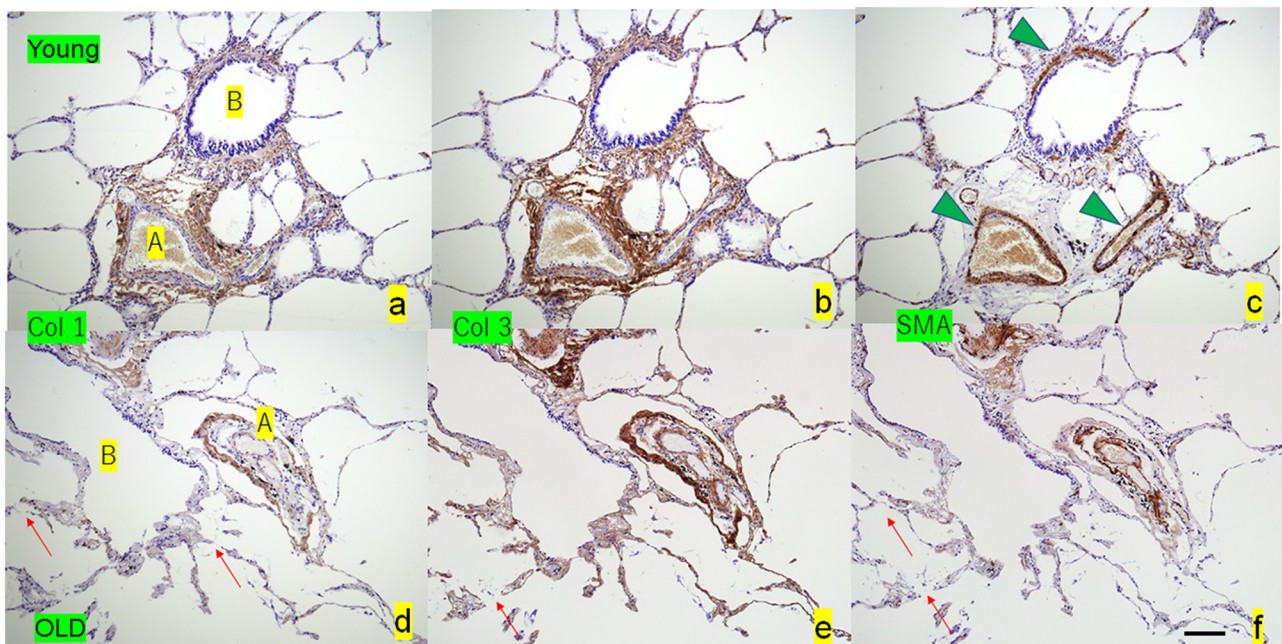

**Fig 6. Comparison of immunostaining of type 1 and type 3 collagen and SMA between young and old lungs.** The young lungs (**a**, **b**, **c**) exhibit small arterioles (A), bronchioles (B), and surrounding alveoli, with conspicuous collagen fibers and thick SMA fibers (arrowheads). Conversely, the old lungs (**d**, **e**, **f**) exhibit dilated bronchioles, thin arterioles with double contours, and disconnected alveoli (arrows), showing weak discontinuous collagen and SMA staining. A, arteriole; B, bronchiole; Col 1, type 1 collagen; Col 3, type 3 collagen; SMA, smooth muscle actin. Scale bar = 200 μm.

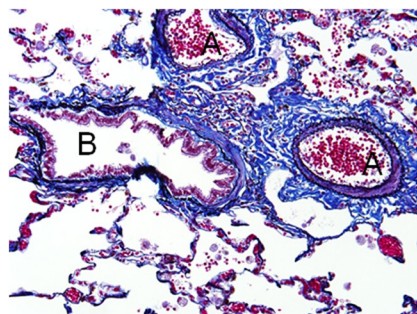

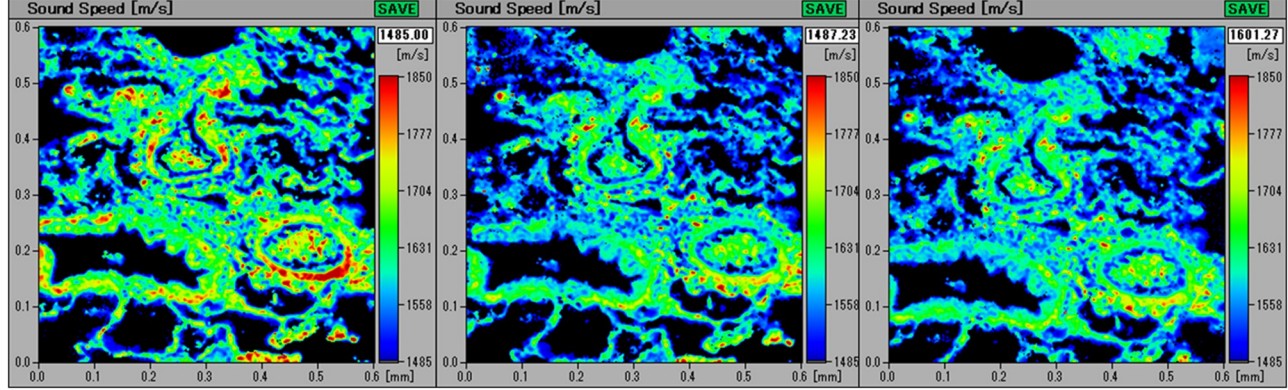

**Fig 7. Alteration in SOS values of the young lung after collagenase digestion.** In the young lung, SOS values of the arteriolar and bronchiolar walls and alveoli show a gradual decline after the collagenase treatment, which can be observed as a change in color indicator. The corresponding LM images of sections stained with Verhoeff's elastic and Masson's trichrome staining are presented in the upper left corner. A, arterioles; B, bronchioles. Scale bar = 200 μm.

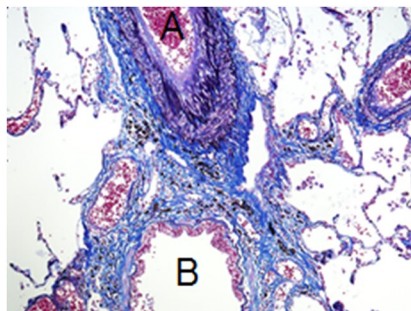

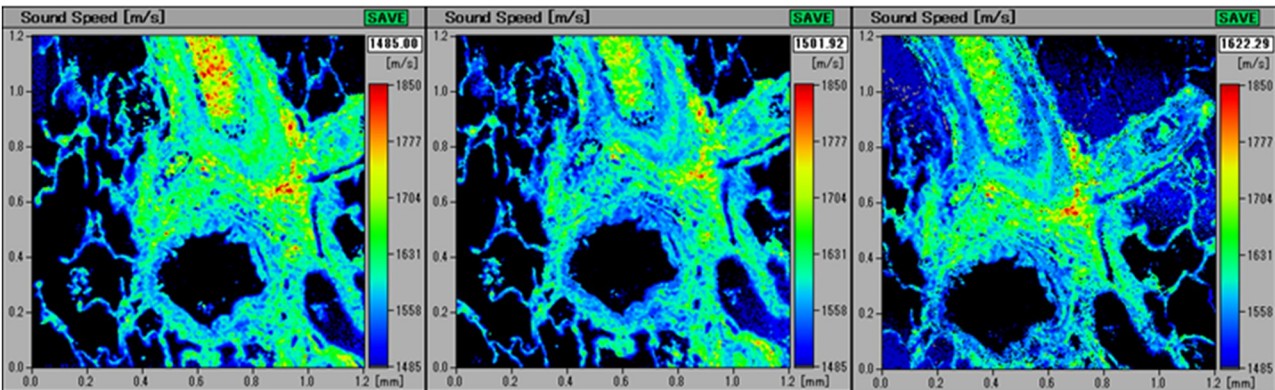

**Fig 8. Alteration in SOS values of the old lung after collagenase digestion.** SOS values of the arteriolar and bronchiolar walls and alveoli in the old lung preserve SOS values after digestion. The blood contents in the arterioles serve as an internal control for collagenase treatment and exhibit a gradual decrease in SOS values after digestion. The corresponding LM images of sections stained with Verhoeff's elastic and Masson's trichrome staining are presented in the upper left corner. A, arterioles; B, bronchioles. Scale bar = 200 μm.

reductions after 1.5 h and 3 h (Fig 9, S3 Table). However, the old group showed no significant decrease in SOS values, except for that observed in the arterioles after 3 h.

### Schematic images of the arterioles, bronchioles, and alveoli associated with aging

Young arterioles consist of thick, smooth muscles and elastic fibers of the medial layer (Fig 10). In contrast, old arterioles are composed of a few smooth muscles with few elastic fibers and focal atheromatous deposits. Therefore, old arterioles lose stiffness with dilatation. Young bronchioles consist of thick, dense connective tissues with few smooth muscles, whereas old bronchioles are composed of loose connective tissues with faint smooth muscles. Consequently, the old bronchioles have lost stiffness and are dilated. Young alveoli are composed of continuous thin walls in regular sizes. In contrast, old alveoli consist of discontinuous walls in larger irregular sizes, resulting in the loss of stiffness of the old alveoli.

### Discussion

In the present study, we examined the changes in SOS by formalin fixation and FFPE methods (Figs 1 and 2). The FFPE sections exhibited significantly higher SOS values than fresh and formalin-fixed sections. However, the three components examined in the study preserved the same order of relative stiffness values between the FFPE and fresh-frozen sections. Even in the FFPE section, arterioles, which were primarily composed of smooth muscle exhibited the highest SOS values; bronchioles, which were chiefly composed of collagen fibers, exhibited

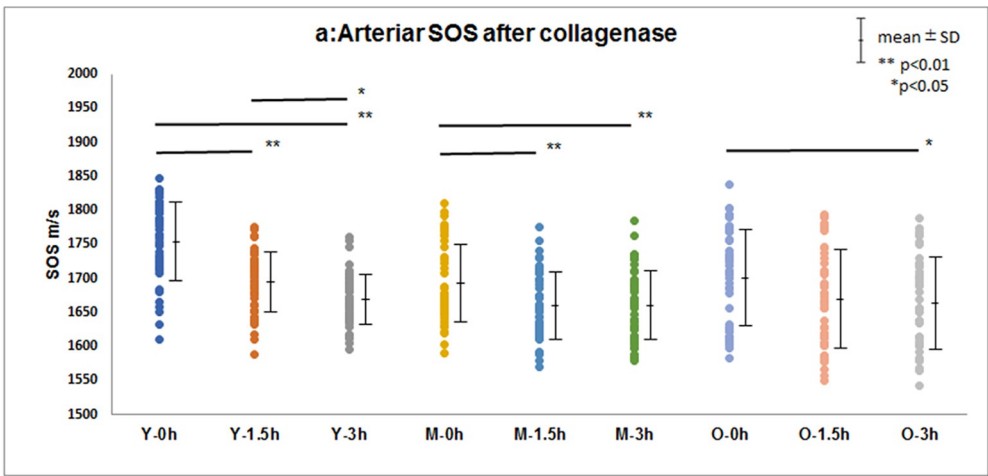

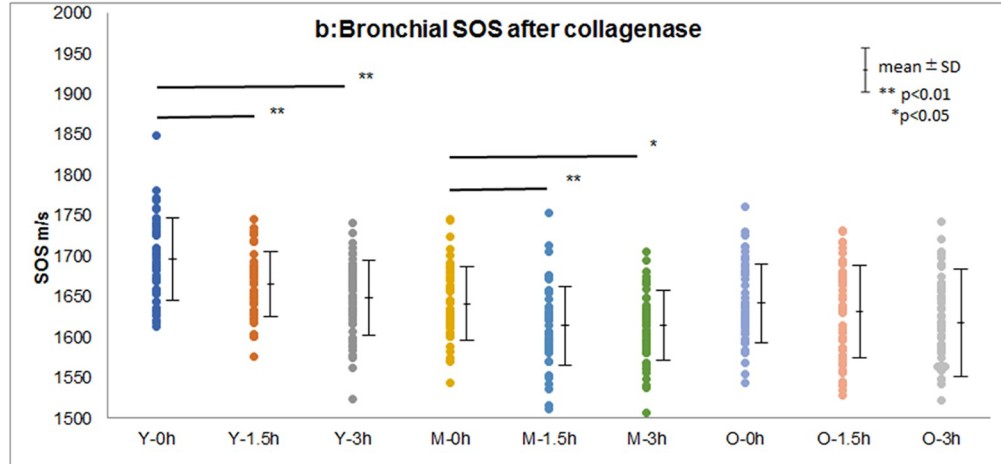

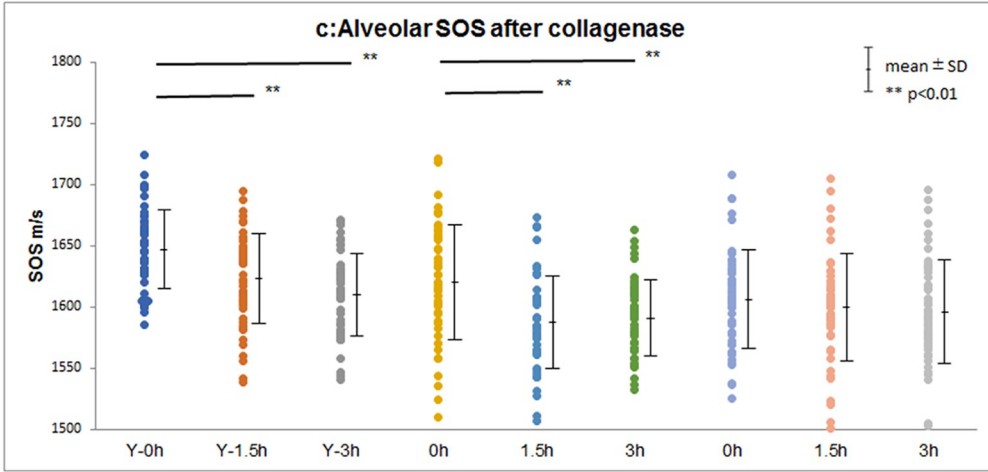

**Fig 9. Change in SOS values over time after collagenase treatment among young, middle-aged, and old groups. (a)** Arteriolar SOS, **(b)** bronchiolar SOS, and **(c)** alveolar SOS. Dot plots of mean SOS values (± standard deviation) are shown. Young and middle-aged lungs show significant reduction in SOS values 1.5 h and 3 h after digestion. However, old lungs show no considerable reduction in SOS values, except for the SOS values of arterioles after 3 h. Y, young; M, middle-aged; O, old.

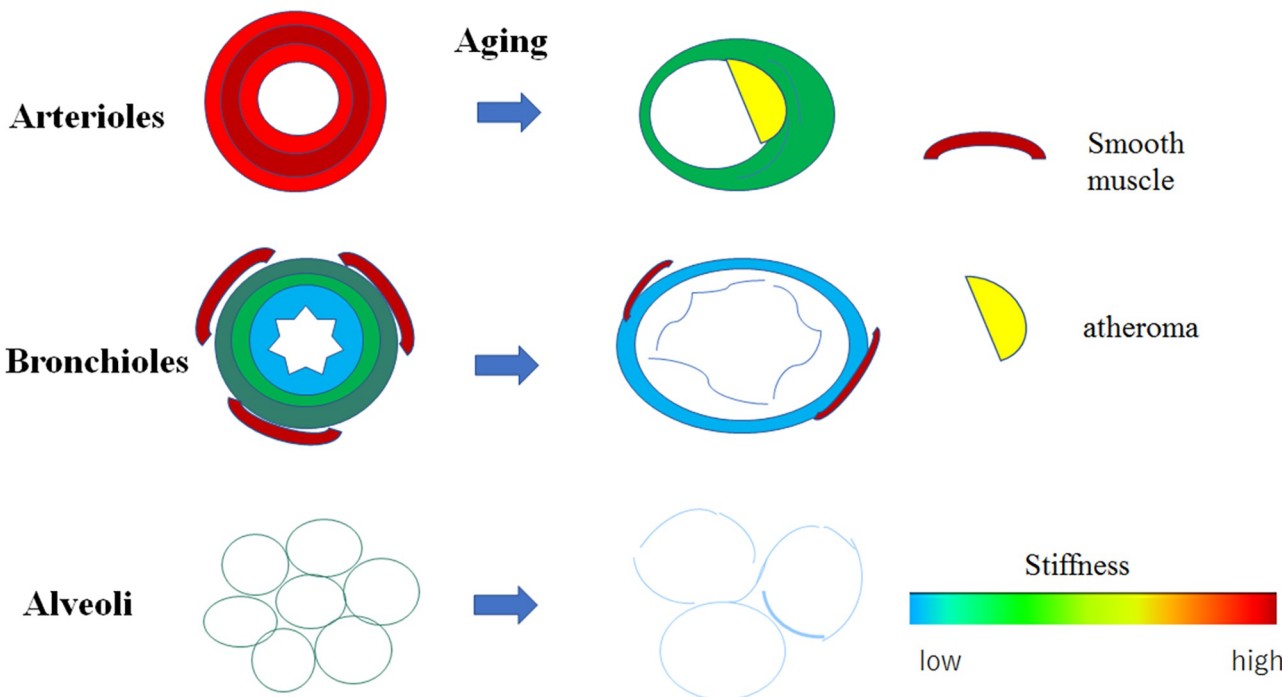

**Fig 10. Schematic images showing alterations associated with aging in the arterioles, bronchioles, and alveoli.** Young arterioles consist of thick, smooth muscles and elastic fibers in the medial layer. Conversely, old arterioles are composed of few smooth muscles and elastic fibers with dilatation and focal atheromatous deposits in the intimal layer. The stiffness of young arterioles is higher than that of the old arterioles. Young bronchioles consist of thick, dense connective tissue with few smooth muscles, whereas old bronchioles are composed of loose connective tissue with sparse smooth muscles. Therefore, the stiffness of old bronchioles is lower than that of the young bronchioles; the old bronchioles are also dilated. Young alveoli are composed of continuous walls with a regular size, whereas old alveoli consist of discontinuous walls in larger irregular sizes, resulting in loss of stiffness.

intermediate SOS values; and alveoli, which were composed of capillaries and thin epithelial cells, exhibited the lowest SOS values.

SOS values of the bronchioles, arterioles, and alveoli gradually declined with aging, indicating that these components lost their stiffness with aging. Decreased stiffness corresponded well to structural alteration. The arteriolar walls of the old lung showed increased collagen fibers and intimal thickness. However, reduced muscles overcame these factors to contribute to declined stiffness. The bronchioles of the old lung had sparse smooth muscles and collagen fibers, which eventually resulted in the loss of stiffness and led to bronchiolar dilatation. The young alveoli consisting of small regular-sized alveoli with a continuous membrane changed to larger irregular-sized structures with a fragile membrane with aging.

Concerning SOS reduction after collagenase digestion, the values significantly declined in the lungs of young and middle-aged groups, whereas those of the old group did not substantially decrease. These results indicated that young and middle-aged lungs were vulnerable to protease digestion, such as inflammatory damage, and reached an irreversible state without repairing ability due to aging.

Various collagens constitute approximately 15% of human lung tissues [12]. Type 1 and type 3 collagen are the most abundant lung collagens, co-distributed in the airways, blood vessels, and interstitium. Type 3 collagen is reportedly increased in the lungs of the old group [13]. The synthesis and degradation rates of collagens are relatively rapid [14]. In adult rats, the turnover rate is approximately 10% per day, and the synthesis rate markedly decreased with aging [15]. Synthesis is active even in old animals, where the bulk of collagens produced are destined to be degraded [15]. In degradation, approximately 30% of the lung collagen is

intracellularly degraded by fibroblasts before the secretion within minutes of its synthesis. The extracellular collagen is broken down more slowly [16]. Highly cross-linked collagen cannot be broken down under normal conditions. This study showed that the young lung produced more collagen, which was easily degraded to turn over rapidly. In contrast, the lung of the old group had little collagen, which is resistant to collagenase to turn over slowly.

To date, a few research studies using SAM have been published for clinical lung pathology. SAM made similar histological images as LM because each pulmonary component had standard SOS values, which varied according to pathological states such as inflammation and neoplasia [11]. SOS in invasive adenocarcinoma was higher than that in adenocarcinoma in situ, and SOS in fibrosis of the usual interstitial pneumonia was higher than that in nonspecific interstitial pneumonia.

Recently, Sicard et al. reported lung stiffness with aging using atomic force microscopy (AFM) [17] observed anatomical variations in stiffness using human donor lungs for transplantation. Airways or bronchioles (200–350 μm diameter) were stiffest. The parenchyma or alveoli were the most compliant. The vessels (<100 μm diameter), including the arterioles and venules, had intermediate stiffness and showed more elasticity in a larger diameter. Comparing young (11–30 years) and older (41–60 years) groups, stiffness of the alveoli and vessels was higher in the older group. The primary culture cells of human smooth muscle cells in the pulmonary artery showed no alteration in mechanical properties at the cellular level but increased traction forces and extracellular matrix deposition with aging. The anatomic variation of stiffness in the present study revealed that the stiffness was highest in arterioles, intermediate in bronchioles, and lowest in alveoli, in contrast to the findings reported by Sicard et al. Venules were discriminated from arterioles and excluded for calculation. Regarding age-related stiffness, all regions decreased with aging in this study. The discrepancy in the findings between the study by Sicard et al. and the present study might be explained by the differences in the methods used, region measurements, and fixation effects. Sicard et al. measured small- and large-sized bronchioles and vessels of the arterioles and venules. In the SAM observation, images were the same as in the LM image. The region of interest was easily and accurately determined.

Many research groups have measured the mechanical properties of the lungs. The elastic modulus of the lung parenchyma varies depending on the technique applied. A recent report compared Young's modulus of the bulk lung tissue using different methods, including AFM microindentation (1.4 ± 0.4 kPa), small amplitude oscillatory shear (3.3 ± 0.5 kPa), uniaxial testing (3.4 ± 0.4 kPa), and cavitation rheology (6.1 ± 1.6 kPa) [18]. These values were within the same magnitude order. Compared with previous modalities, SAM showing the alveolar SOS was approximately 1600–1650 m/s, corresponding to 2.56–2.72 GPa of Young's bulk modulus. Young's modulus ($E$) is obtained using the following formula: $E = c^2 \times \rho$ where $c$ is SOS and $\rho$ is density. $\rho$ is about $10^3$ kg/m$^3$.

Young's bulk modulus of the pulmonary tissue obtained using SOS was relatively higher in GPa level than those measured using other methods. In general, the wave speed in liquid is the square root of a measure of stiffness divided by density. "Compression-type" sound is used to detect the target in the SAM system, and the SOS is usually faster in solid than liquid. Water accounts for 83.74% of the lung [19]; therefore, sound waves are transmitted through solid materials in water. Tissue density is slightly lower in lipids, approximately 0.9 g/cm$^3$, and a little higher in proteins, approximately 1.1 g/cm$^3$ than water, 1.0 g/cm$^3$, indicating that SOS is positively related to the stiffness. However, SOS values are similar to those of other soft tissue organs, such as the connective tissue, i.e., 1545 m/s (IT'IS Foundation, https://itis.swiss/virtual-population/tissue-properties/database/acoustic-properties/speed-of-sound/). An air-filled lung shows SOS values of 949.3 m/s between water (1485 m/s) and air (343 m/s) values.

In our SAM observation, tissue sections were fastened on the glass slide in a dry state and soaked in fixatives. Therefore, the sound waves were transmitted through the solid in liquid, whereas in other methods, samples might be treated as liquid because it is unfixed and freely floating in the water or air. SOS values reflect dry materials as dry fish are far stiffer similar to bones compared with fresh fish.

Sicard et al. [17] reported that the Young modulus of the alveolar wall tissue is 1–3 kPa, which is remarkably lower than that of water (2 GPa) and airways (14–17 kPa) because lipid contents of surfactant produced by alveolar type II cells were located on the alveolar walls, which reduce the surface tension to expand alveolar walls in a free-moving state. Alveolar walls are composed of expansible fiber proteins of collagen and elastin as well as surfactants to exhibit compliance and distensibility in a fresh state. However, this study showed relatively high SOS values using fixed lung tissues corresponding to 2.56–2.72 GPa higher than that of water in Young modulus. Surfactant lipids were already removed in the tissue processing, and proteins, including the blood, were fixed to increase the stiffness.

Recently, a novel technique, ultrasound-surface-wave-elastography, has been reported to allow for noninvasive evaluation of lung stiffness in living individuals [20,21]. In this method, low-frequency harmonic vibration is applied on the chest skin in the intercostal space, and the lung surface wave velocity is measured from the nearby intercostal space. This safe method aids in assessing interstitial lung fibrosis but has certain disadvantages because subpleural emphysema or thick subcutaneous can delay wave propagation in reaching the lung. Pleural fibrosis and fluid may interfere with the correct evaluation of lung stiffness. It is difficult to discriminate the structures causing lung stiffness.

The disappearance of supporting peri-bronchial collagens and smooth muscles mechanically influenced the weakness of bronchioles. A recent report indicated that the number of airways smaller than 2.0 mm in diameter but not of airways larger than 2.5 mm decreased with aging, i.e., between 30 and 80 years [22]. The disappearance of small bronchioles may result in the loss of supporting components.

Concerning the alveoli, fiber splitting and collagenase resistance indicate that cross-linking and slow turnover of collagens as well as reduced elasticity cause alveolar expansion with aging. In an immunohistochemical study, type 3 collagen increased in elderly alveolar walls compared with young ones (Fig 6). Conversely, elastic fibers along the alveolar walls have reported no notable changes with aging [13]. Type 1 collagen is tightly packed and form fibers of excellent tensile strength, whereas type 3 collagen usually exists in fibrils of smaller diameter than type 1 [23]. Type 3 collagen allows multidirectional flexibility and increases in early active fibrosis such as that observed in pneumonia [23]. Small-diameter collagen with splitting may reduce SOS, namely, decreased stiffness resulting in alveolar enlargement.

Results of decreased SOS of lung parenchymal components are contrary to typical findings that older lungs become stiffer and lose their elasticity. The cause of this discrepancy is considered as follows. First, average old lungs accepted additional pathological changes such as various inflammatory stimuli and increase stiffness. However, old lungs without further changes show non-stiff characteristics, such as a cotton-candy lung of end-stage emphysema [24]. The selected lung portions for observation contained no inflammatory or fibrous lesions, which represented healthy aging. However, the usual old lungs suffered from many environmental factors that may result in fibrosis. In chronic obstructive pulmonary disease, terminal and respiratory bronchioles show inflammation or fibrosis, and arteries exhibit intimal thickening with smooth muscle proliferation [25]. Second, the amount of protein content in the framework affected the SOS values of each structure, with collagen and elastic fibers and smooth muscles as the main components. The framework of young lungs is more robust than that of older lungs. This study evaluated no airspace and wall inflammatory lesions. Therefore, the

normal framework gradually weakens with aging. Third, the stiffness of the peribronchiolar portions did not represent that of the whole lung. The pleura, subpleural alveoli, and central bronchioles may contribute to the entire lung stiffness.

The SAM observation has several limitations that should be acknowledged. First, the sample bias may be present to evaluate the normal aging process. Archives of pathological specimens were selected due to sufficient remnants of intact parts away from the lesion. However, the individual case had particular lesions or diseases in the background, so that the small number of samples may not represent typical structural and generation alterations. Second, reaction to collagenase treatment may not reflect the general response against proteases. We evaluated pepsin, collagenases, and actinase for protease digestion. Pepsin uses acidic buffers for digestion, which might influence the elasticity. Collagenase and actinase worked in a neutral pH solution. However, control of actinase and some collagenases, robust nonspecific proteases, resulted in a nonsignificant difference among the samples. We found that type 3 collagenase with lower nonspecific proteolytic activity revealed differences among the samples.

The SAM method may be used in pathology practice to investigate various tissues in the future. To discriminate and evaluate pulmonary fibrosis, the area and degree of fibrosis are countable using SOS values. The susceptibility estimates the chronicity to protease digestion. Fresh fibrosis is vulnerable to digestion, whereas chronic fibrosis is resistant. If proper substrate-specific enzymes are found, the contribution of each substrate to stiffness will be elucidated.

## Conclusion

SOS values in the bronchioles, the arterioles, and the alveoli decreased with aging, indicating a reduction in their stiffness. SOS alteration corresponded well to structural changes with aging: loss of muscular and elastic components of the arterioles, bronchioles, and alveoli to be dilated. After the collagenase digestion, responsive alteration showed that young and middle-aged lungs significantly reduced in all parts. Conversely, old lungs were resistant to collagenase digestion to maintain an actual state that had already suffered protease damage.

## Supporting information

**S1 Table. One-way ANOVA for SOS values with different fixation methods.**
(DOCX)

**S2 Table. One-way ANOVA for SOS values in different age groups.**
(DOCX)

**S3 Table. One-way ANOVA for collagenase effects on SOS values indifferent age groups and different durations.**
(DOCX)

## Acknowledgments

The author thanks T. Moriki, Y. Egawa, M. Fujie, Y. Kawabata, and N. Suzuki for preparing the histological samples; Dr. K. Kobayashi (Honda Electronics) for his technical support and advice with SAM; and Enago (www.enago.jp) for the English language review.

## Author Contributions

**Conceptualization:** Katsutoshi Miura.

**Formal analysis:** Katsutoshi Miura.

**Funding acquisition:** Katsutoshi Miura.

**Investigation:** Katsutoshi Miura.

**Methodology:** Katsutoshi Miura.

**Project administration:** Katsutoshi Miura.

**Visualization:** Katsutoshi Miura.

**Writing – original draft:** Katsutoshi Miura.

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
