## [Decision Letter · Decision Letter 0]

8 Nov 2021

PONE-D-21-28797Stiffness reduction and collagenase resistance of an aging lung measured using scanning acoustic microscopyPLOS ONE

Dear Dr. Miura,

Thank you for submitting your manuscript to PLOS ONE. After careful consideration, we feel that it has merit but does not fully meet PLOS ONE’s publication criteria as it currently stands. Therefore, we invite you to submit a revised version of the manuscript that addresses the points raised during the review process.

 The reviewers raise a number of serious issues. Each of these should be addressed carefully. These include, but are not limited to, the need for additional control samples, the title and focus of the work, and apparent large disagreements with the literature on stiffness values. Further experimental work may need to be presented to satisfy some of these points. 

We look forward to receiving your revised manuscript.

Kind regards,

Richard G. Haverkamp, PhD

Academic Editor

PLOS ONE

Journal Requirements:

Reviewers' comments:

Reviewer's Responses to Questions

**Comments to the Author**

1. Is the manuscript technically sound, and do the data support the conclusions?

Reviewer #1: Partly

Reviewer #2: Yes

2. Has the statistical analysis been performed appropriately and rigorously? 

Reviewer #1: I Don't Know

Reviewer #2: I Don't Know

3. Have the authors made all data underlying the findings in their manuscript fully available?

Reviewer #1: Yes

Reviewer #2: Yes

4. Is the manuscript presented in an intelligible fashion and written in standard English?

Reviewer #1: Yes

Reviewer #2: Yes

5. Review Comments to the Author

Reviewer #1: This article is presenting the characterization of mechanical properties of human lung tissue using scanning acoustic microscopy. The author firstly tested the effect of several fixation protocols on SOS values. Using 3 different aging groups, the author has compared the stiffness of lung components and evaluated the aging alteration on structure and mechanical properties. Finally, the author has demonstrated the reduction of SOS values after collagenase digestion of lung tissue.

According to all major comments presented below, this article, by this figure presentation, the methods used, the data presented, and the conclusion made are not enough precise and strongly supported to be accepted.

Major Comments:

1- Considering the capabilities of the current basic software to edit article figure, the author should do an effort to present a better figure quality. Too many obvious “screenshots” and cropping in the figures.

2- The author used some lung tissue from a same patient to compare different fixation methods.

- First, the author doesn’t clearly explain the interest of this experiment in the article. Why is the fixation step required to analyze lung tissue stiffness using SAM?

- Why did the author do several times of fixation using the same tissue slice? The fact to stop the fixation, analyze the sample into water and fix it again for the longer time has a real interest?

- If the goal was to compare fresh frozen fixed tissue to FFPE sections and show that FFPE sections were usable for SAM experiment, it was more pertinent to analyze “fresh/frozen no fixation” tissue and then fix it with a protocol close to the FFPE one, like 10% formalin overnight, and finally compare with FFPE section.

3- Over the comparison between different fixation protocols, the fact to fix tissue to characterize its stiffness seems aberrant. The author explains himself in the discussion that “FFPE mainly affected the protein bridging” (lines 362-363). One negative control experiment with a tissue section analyzed by SAM with no fixation is strongly necessary to prove that FFPE processing doesn’t affect/change tissue stiffness.

4- The author explained that “mean SOS values were calculated from the values of at least five different points of each lung component”. The author should be more specific. Considering bronchiole or arteriole, these points were taken close to each other or all around the wall? With a square frame comprising 300x300 points per SOS images, the author could definitively add more values to have more representative mean SOS values for each lung component.

5- Considering the resolution of the SAM (1pixel equal to 3-4 µm), any stiffness difference has been observed by the author for the different cell layers of bronchiole and arteriole?

6- How does the fixation process affect the collagenase digestion? What is the scientific interest in this study to compare 1.5h to 3h? Why the nonremarkable reduction of collagen for elderly group is not due to the high cross-linking of collagen fiber after fixation?

7- In Figure 3a, in SAM images for all aging groups, some areas in alveoli tissue and inside airway lumen show higher elasticity than the substrate (blue background vs black background).

8- It would be interesting to have a difference of color or dot shape into the graphs in particular in Figure 3C to identify the values of each patients (from Table1).

9- In Figure 7, the author gave a schematic image of lung constituents associated with ageing. Focal atheromatous deposits have been included in the schematic but there are any explanations anywhere in the article about this addition. The yellow color code is supposed to give information about this stiffness but any literature reference or experimental data in the article are supporting this result.

10- The author compares his lung tissue stiffness results to four different other techniques present in the literature. All of them are reporting elastic values between 1 and 6 kPa. The author reports elastic values around 2.56-2.76 GPa, so 10^6 times higher. This is aberrant… The tissue and organs have all their own stiffness range (DOI: 10.1080/15476278.2015.1019687). The author should show at least that this same range exists using SAM to give some possible validations to these aberrant elastic values.

Minor Comments:

1- In Figure1, this is written that the tissue section is from “mouse lung” (line. 190). In sample preparation paragraph, it is explained that the experiment has been performed on human samples.

2- To get some clarification, each point in the graphs is the mean SOS value of one bronchiole, arteriole, or alveoli region?

3- How did the author make the difference between arteriole and venule into lung tissue?

4- In Figure 5b, other SOS and staining pictures should be used to have similar image and size of airway and arteriole to compare.

5- Not “Turkey-Kramer test” but Tukey-Kramer test.

6- The sentence on lines 333 and 334 must be re-written.

Reviewer #2: Title: Stiffness reduction and collagenase resistance of an aging lung measured using scanning acoustic microscopy

This is a very interesting article presenting the efficiency of scanning acoustic microscoy to analyse the histo-mechanical proerties. I enjoyed going through it, however, I would like to raise the following observations, and find them critical to decide accepting the article for publication:

I thing the title is irrelevant for the following reasons:

1. Stiffness reduction implies a comparison with a known standart values of the involved stiffness moduli of the specific sample

2. „An aging lung“: indicates a measurement of only one sample, and even in such a case it is a tissue sample of micrometer scale and is not a representative for even a lung sample

In the Abstract:

- „Formalin-fixed, paraffinembedded (FFPE) sections for LM were used for comparison, showing no significant difference of SOS values from almost fresh samples“ misleads to that the light microscope is the tool used for SOS comparisons.

- „SOS values of all the components were significantly reduced in young and middle-aged groups but not in the elderly group. Protease damage was accumulated in old lungs without regeneration and loss of elastic components“ A reduced SOS corresponds to increased stiffness rathher than loss of stiffness !!!!

MAS-coated glass slides ? Are they specific glass that provide electrostatic attraction ?

distilled water ? The coupling fluid should be isotonic solution to avoid osmosis !! imbibtion of water could lead to changes in the mechanical properties

The followed fixation protocol is not refered to the source references

A one-way and two ways ANOVA and Turkey–Kramer post hoc tests ; Are not clear for me.

The mass density of the tissues is assumed fixed, how do you verify insuring that no changes in density is taking place with aging ?

-L 448, the sound waves were transmitted through the solid in liquid, but glass should be highly reflective!!

-It is not clear to me how it is justified that the variations observed in the velocity of ultrasound is due to aging rather than being an effect of the fixation

- L 323 Dot blots of SOS values with mean and SD were shown. I think it is meant Plots and not blots.

How can you justify the high standard deviation from the mean values in most cases, in figure 6, as an example? .

In line 171, there is a spelling mistake Lab view instead of LavView

6. PLOS authors have the option to publish the peer review history of their article (what does this mean?). If published, this will include your full peer review and any attached files.

Reviewer #1: No

Reviewer #2: No

---

## [Author Response · Author response to Decision Letter 0]

20 Jan 2022

Response to Reviewers

Reviewer #1: This article is presenting the characterization of mechanical properties of human lung tissue using scanning acoustic microscopy. The author firstly tested the effect of several fixation protocols on SOS values. Using 3 different aging groups, the author has compared the stiffness of lung components and evaluated the aging alteration on structure and mechanical properties. Finally, the author has demonstrated the reduction of SOS values after collagenase digestion of lung tissue.

According to all major comments presented below, this article, by this figure presentation, the methods used, the data presented, and the conclusion made are not enough precise and strongly supported to be accepted.

Major Comments:

1- Considering the capabilities of the current basic software to edit article figure, the author should do an effort to present a better figure quality. Too many obvious "screenshots" and cropping in the figures.

Answer: We thank the reviewer for this suggestion. Cropping the images showing SOS values was necessary to create figures using images captured from the computer screen. We have increased the resolution of figures to present better-quality figures using the Photoshop software; the revised figures have been submitted with the revised manuscript. 

2- The author used some lung tissue from a same patient to compare different fixation methods.

- First, the author doesn't clearly explain the interest of this experiment in the article. Why is the fixation step required to analyze lung tissue stiffness using SAM?

Answer: We considered that fresh frozen tissues without fixation would not be suitable for observation using SAM because water-soluble materials are lost from the tissue. Cells are exposed to low osmotic pressure, and residual enzymes break down materials. In our experience, light microscopic images show that free cells in fresh specimens degenerate, which distinguish these specimens from fixed specimens. For clinical specimens, avoiding infectious material is also essential. Therefore, we have assumed that fixation would be necessary to examine clinical tissue samples that remain stable. Therefore, to detect the effects of fixation on SOS values, we evaluated the changes in SOS after fixation and embedding.

- Why did the author do several times of fixation using the same tissue slice? The fact to stop the fixation, analyze the sample into water and fix it again for the longer time has a real interest?

Answer: Comparison of different sections or areas is not suitable because of variations in components and local thickness and local fixation bias. We aimed to reduce tissue bias and examine the effect of only the fixation method by observing the same area of the same section. 

- If the goal was to compare fresh frozen fixed tissue to FFPE sections and show that FFPE sections were usable for SAM experiment, it was more pertinent to analyze "fresh/frozen no fixation" tissue and then fix it with a protocol close to the FFPE one, like 10% formalin overnight, and finally compare with FFPE section.

Answer: We agree with the reviewer's comments that the goal of this examination was to evaluate the quality of the FFPE sections. We compared fresh/frozen sections without fixation with sections fixed with formalin for 24 h and FFPE sections. The results are presented in revised Fig. 1. To minimize the bias in sections, we prepared several sections from nearby portions and compared fresh-frozen, formalin-fixed, and FFPE sections. The fresh-frozen sections exhibited the lowest SOS values, and the FFPE sections always displaｙed significantly higher SOS values than the fresh-frozen sections. We have prepared new Fig 1 and 2 and revised the Methods and Results sections regarding the effects of fixation on SOS values (Methods section, lines 91–107; Results section, lines 194–203; Fig 1, lines 205–215; Fig 2, lines 226–230).

3- Over the comparison between different fixation protocols, the fact to fix tissue to characterize its stiffness seems aberrant. The author explains himself in the discussion that "FFPE mainly affected the protein bridging" (lines 362-363). One negative control experiment with a tissue section analyzed by SAM with no fixation is strongly necessary to prove that FFPE processing doesn't affect/change tissue stiffness.

Answer: We thank the reviewer for their suggestion. We prepared fresh-frozen sections without fixation, as shown in the revised Fig 1, to compare with FFPE sections and to ensure that FFPE processing did not affect tissue stiffness. We found that the FFPE sections exhibited significantly higher SOS values than the fresh-frozen sections. However, even in FFPE sections, the SOS values were highest in the arterioles, intermediate in the bronchioles, and lowest in the alveoli. FFPE tissues became stiffer but retained relative stiffness reflecting fresh samples.

4- The author explained that "mean SOS values were calculated from the values of at least five different points of each lung component". The author should be more specific. Considering bronchiole or arteriole, these points were taken close to each other or all around the wall? With a square frame comprising 300x300 points per SOS images, the author could definitively add more values to have more representative mean SOS values for each lung component.

Answer: Regarding the bronchioles or arterioles, several points from all around the wall were used for analysis. Specifically, we placed a small spot on each region of interest on the screen and determined the SOS value. To determine the mean SOS value of each component, we selected at least five different images and measured the SOS values of each component (lines 131–134).

5- Considering the resolution of the SAM (1pixel equal to 3-4 µm), any stiffness difference has been observed by the author for the different cell layers of bronchiole and arteriole?

Answer: We thank the reviewer for their inquiry. The bronchiolar walls are composed of smooth muscle and collagen fibers, and the arteriolar walls are composed of intima, media, and adventitia. The limited resolution of the transducer and the thick walls with the small width of the bronchioles, arterioles, and alveoli made it challenging to differentiate the precise borders among these layers. Moreover, these layered structures become increasingly obscure with aging. 

6- How does the fixation process affect the collagenase digestion? What is the scientific interest in this study to compare 1.5h to 3h? Why the nonremarkable reduction of collagen for elderly group is not due to the high cross-linking of collagen fiber after fixation?

Answer: Each protein has a definite ratio of amino acids. Formalin fixation creates cross-links among proteins such as arginine, lysine, serine, and tyrosine residues, forming reactive complexes. (Eltoum et al. J Histotechnol 2001;24;173-190). Collagens are composed of repeating Xaa-Yaa-Gly sequences, where Xaa and Yaa can be any amino acid. Bacterial collagenase breaks down the peptide bonds of Yaa-Gly. Due to protein modifications, the collagenase digestion of fresh tissues is slightly more effective than fixed tissues, including cross-links among proteins. However, all fixed tissues are digestible by breaking the peptide bonds using the same method. Comparing the SOS values at 1.5 h and 3 h after digestion allowed us to clarify the time course of digestion effects. The alteration in SOS values revealed the speed of digestion. An incubation time that is too short does not cause a significant reduction in the SOS value. At least two time points are necessary to determine the proper incubation time for the comparison of digestion speed. Protein cross-links accumulate with aging according to the cross-linkage theory of aging proposed by Johan Björkstein in 1942. Although protein cross-links are more developed in the elderly, the effects of formalin fixation are stable because formalin preserves structures regardless of the tissue age. The effect of aging on cross-links may interfere with formalin fixation, which causes increased collagenase effects.

7- In Figure 3a, in SAM images for all aging groups, some areas in alveoli tissue and inside airway lumen show higher elasticity than the substrate (blue background vs black background).

Answer: The areas in blue color correspond to similar basal values of SOS, which is the same as water. The variation in slide thickness and difference in distance from the microscope length lead to these variations; the currently available software is unable to address this issue.

8- It would be interesting to have a difference of color or dot shape into the graphs in particular in Figure 3C to identify the values of each patients (from Table1).

Answer: We thank the reviewer for their suggestion. However, the currently available software does not have the ability to plot the SOS value for each patient in a different color. The use of too many dots with different colors may hinder the observation of distinct colors.

9- In Figure 7, the author gave a schematic image of lung constituents associated with ageing. Focal atheromatous deposits have been included in the schematic but there are any explanations anywhere in the article about this addition. The yellow color code is supposed to give information about this stiffness but any literature reference or experimental data in the article are supporting this result.

Answer: We thank the reviewer for their comment. In Fig. 3a and 5b, the atheromatous portions in the intima exhibited high SOS values. In the presence of calcification, the ultrasound reflects irregularly and interferes with imaging. We have added this explanation in the legend for Fig. 3a as follows:

Fig 3. SOS and LM images of the lung in different age groups

(a) SOS images of the young, middle-aged, and old lungs. SOS values of the bronchioles, arterioles, and alveoli are high in the young group, which gradually become lower with the increasing age of the lungs, except for the intimal atheromatous region forming a ring shape in the arteries of the old group, which exhibits high SOS values indicated as yellow.

10- The author compares his lung tissue stiffness results to four different other techniques present in the literature. All of them are reporting elastic values between 1 and 6 kPa. The author reports elastic values around 2.56-2.76 GPa, so 10^6 times higher. This is aberrant… The tissue and organs have all their own stiffness range (DOI: 10.1080/15476278.2015.1019687). The author should show at least that this same range exists using SAM to give some possible validations to these aberrant elastic values.

Answer: We thank the reviewer for their insightful inquiry. Wet tissues such as organs can change volume against a pushing force. Water in organs can escape away from the region exposed to the pushing force. During SAM observation, ultrasound velocity through the solid materials is measured to compare with a water velocity of 1485 m/s. For example, dry fish is far stiffer, akin to bones, than fresh fish. Denser tissues are stiffer and exhibit higher SOS values, which reflect materials in a dry state. This aspect is the main reason for the tremendous discrepancy in stiffness values. We have added this explanation to the revised manuscript (lines 436–440).

Minor Comments:

1- In Figure1, this is written that the tissue section is from "mouse lung" (line. 190). In sample preparation paragraph, it is explained that the experiment has been performed on human samples.

Answer: We thank the reviewer for their careful observation and apologize for the grave error. We had indeed used mouse lung tissue in preliminary studies to compare fixation effects; those results have been deleted. We have therefore corrected the wording in the revised manuscript. 

2- To get some clarification, each point in the graphs is the mean SOS value of one bronchiole, arteriole, or alveoli region?

Answer: We thank the reviewer for giving the opportunity to explain. In the graph, each point represents the SOS value calculated for each region in bronchioles, arterioles, or alveoli. The total numbers of regions that were plotted were comparable among the patients. 

3- How did the author make the difference between arteriole and venule into lung tissue?

Answer: Arterioles have a thick medial layer with a round shape, while venules have a thin medial layer with dilatation. Moreover, arterioles run along the bronchioles, whereas venules do not pass through the bronchioles.

4- In Figure 5b, other SOS and staining pictures should be used to have similar image and size of airway and arteriole to compare.

Answer: We thank the reviewer for their inquiry. The old lungs show dilated bronchioles and arterioles compared with the young lungs. Therefore, it is challenging to display all three components (bronchioles, arterioles, and alveoli) in the same image with the same magnification, such as that can be done in the young lungs. However, we have revised Fig 5b to a new image showing all components, although the magnification is slightly lower than that of the image in Fig. 5a. Moreover, we now show blood contents as an internal control for digestion.

5- Not "Turkey-Kramer test" but Tukey-Kramer test.

Answer: We thank the reviewer and apologize for the oversight. We have corrected the error.

6- The sentence on lines 333 and 334 must be re-written.

Answer: These sentences have been deleted.

Reviewer #2: Title: Stiffness reduction and collagenase resistance of an aging lung measured using scanning acoustic microscopy

This is a very interesting article presenting the efficiency of scanning acoustic microscopy to analyse the histo-mechanical properties. I enjoyed going through it, however, I would like to raise the following observations, and find them critical to decide accepting the article for publication:

I think the title is irrelevant for the following reasons:

1. Stiffness reduction implies a comparison with a known standard values of the involved stiffness moduli of the specific sample

2. "An aging lung ": indicates a measurement of only one sample, and even in such a case it is a tissue sample of micrometer scale and is not a representative for even a lung sample

Answer: We thank the reviewer for their essential insight. We believe that the title correctly reflects the context of the study. Speed of sound through tissues is one of the modules indicating stiffness. Regarding comparison with other methods, we kindly refer the reviewer to our response to inquiry #10 of reviewer 1. Additionally, we have changed "an aging lung" in the title to "aging lungs" to generalize our findings.

In the Abstract:

- "Formalin-fixed, paraffin embedded (FFPE) sections for LM were used for comparison, showing no significant difference of SOS values from almost fresh samples "misleads to that the light microscope is the tool used for SOS comparisons.

Answer: We thank the reviewer for their recommendation. We have deleted the sentence referring to LM for SOS evaluation in the revised manuscript. As suggested by reviewer 1, we repeated the experiments comparing fresh-frozen sections with FFPE sections. Our analyses revealed that the FFPE sections exhibited significantly higher SOS values than the fresh-frozen sections, as presented in the revised Fig 1 and 2. We have therefore updated the conclusion. However, the FFPE tissues retained relative stiffness reflecting fresh samples. 

- "SOS values of all the components were significantly reduced in young and middle-aged groups but not in the elderly group. Protease damage was accumulated in old lungs without regeneration and loss of elastic components ". A reduced SOS corresponds to increased stiffness rather than loss of stiffness !!!!

Answer: We thank the reviewer for their opinion. However, a decrease in SOS indicates reduced stiffness, which is supported by the reduction of tissue components such as smooth muscle, collagens, and elastic fibers. This change happens in healthy lungs. In disease states, inflammation induces tissue fibrosis, which leads to tissue stiffness.

MAS-coated glass slides ? Are they specific glass that provide electrostatic attraction ? 

Answer: That is correct; these slides were used to mount the sections on glass slides.

distilled water? The coupling fluid should be isotonic solution to avoid osmosis !! imbibtion of water could lead to changes in the mechanical properties

Answer: We thank the reviewer for their inquiry. All tissues were fixed before observation; therefore, water could go in and out freely. The difference in osmotic pressure has no effect on mechanical properties. In the new experiments using fresh tissue sections, we used physiologic saline solution as the coupling fluid.

The followed fixation protocol is not refered to the source references

Answer: The fixation protocol used in the present study is a standard protocol used for the preparation of pathology specimens. We have added the post-fixation protocol in the Methods section in the revised manuscript (lines 105–107).

A one-way and two ways ANOVA and Turkey–Kramer post hoc tests ; Are not clear for me. 

Answer: We thank the reviewer for their inquiry. We have revised the Statistical Analyses section to clarify the methods used (lines 171–191).

The mass density of the tissues is assumed fixed, how do you verify insuring that no changes in density is taking place with aging?

Answer: We believe that the mass density of the lungs decreases with aging in healthy lungs because of the dilation of air space and decreased regenerative activity.

-L 448, the sound waves were transmitted through the solid in liquid, but glass should be highly reflective!!

Answer: Tissue components are made of organic compounds, although glass contains minerals that exhibit exceedingly high SOS. Sound reflects at the border of materials with high SOS values.

-It is not clear to me how it is justified that the variations observed in the velocity of ultrasound is due to aging rather than being an effect of the fixation

Answer: Regarding the effects of fixation, as we indicated in our response to reviewer 1, the FFPE tissues became stiffer than the fresh-frozen tissues but retained relative stiffness reflecting fresh samples. The significance of the variations among different age groups was stronger than that of the variations within the same age groups (S2 Table). This finding indicates that the significant difference in SOS values was a result of aging and not fixation. 

- L 323 Dot blots of SOS values with mean and SD were shown. I think it is meant Plots and not blots.

Answer: We thank the reviewer for their recommendation. We have corrected the sentence accordingly. 

How can you justify the high standard deviation from the mean values in most cases, in figure 6, as an example? .

Answer: Each dot corresponds to one point value of SOS. Because various structures were present in arterioles, bronchioles, and alveoli, it is natural that loose or dense components in a 2 × 2-µm2 area of 300 × 300 points generate multiple values. 

In line 171, there is a spelling mistake Lab view instead of LavView

Answer: We thank the reviewer for their observation and apologize for the oversight. The spelling mistake has been corrected.

---

## [Editor Report · Decision Letter 1]

31 Jan 2022

Stiffness reduction and collagenase resistance of aging lungs measured using scanning acoustic microscopy

PONE-D-21-28797R1

Dear Dr. Miura,

We’re pleased to inform you that your manuscript has been judged scientifically suitable for publication and will be formally accepted for publication once it meets all outstanding technical requirements.

Kind regards,

Richard G. Haverkamp, PhD

Academic Editor

PLOS ONE
---

## [Editor Report · Acceptance letter]

9 Feb 2022

PONE-D-21-28797R1 

Stiffness reduction and collagenase resistance of aging lungs measured using scanning acoustic microscopy 

Dear Dr. Miura:

I'm pleased to inform you that your manuscript has been deemed suitable for publication in PLOS ONE. Congratulations! Your manuscript is now with our production department. 

Kind regards, 

on behalf of

Professor Richard G. Haverkamp 

Academic Editor

PLOS ONE